# Disentangled Knowledge Tracing for Alleviating Cognitive Bias

## Abstract

In the realm of Intelligent Tutoring System (ITS), the accurate assessment of students' knowledge states through Knowledge Tracing (KT) is crucial for personalized learning. However, due to data bias, *i.e.*, the unbalanced distribution of question groups (*e.g.*, concepts), conventional KT models are plagued by cognitive bias, which tends to result in cognitive underload for overperformers and cognitive overload for underperformers. More seriously, this bias is amplified with the exercise recommendations by ITS. After delving into the causal relations in the KT models, we identify the main cause as the confounder effect of students' historical correct rate distribution over question groups on the student representation and prediction score. Towards this end, we propose a Disentangled Knowledge Tracing (DisKT) model, which separately models students' familiar and unfamiliar abilities based on causal effects and eliminates the impact of the confounder in student representation within the model. Additionally, to shield the contradictory psychology (*e.g.*, guessing and mistaking) in the students' biased data, DisKT introduces a contradiction attention mechanism. Furthermore, DisKT enhances the interpretability of the model predictions by integrating a variant of Item Response Theory. Experimental results on 11 benchmarks and 3 synthesized datasets with different bias strengths demonstrate that DisKT significantly alleviates cognitive bias and outperforms 14 baselines in evaluation accuracy.

## Keywords

Knowledge Tracing, Educational Data Mining

## 1 Introduction

In recent years, especially with the explosion of large language models (*e.g.*, GPT-4o), AI for Education has received widespread attention [3, 14, 61, 71]. Intelligent Tutoring System (ITS), as a component of this field, has also seen rapid development [29, 69]. The success of ITS lies in its ability to recognize each student's knowledge state and recommend personalized learning resources (*e.g.*, exercises) based on the large-scale learning data obtained from online learning environments [32]. *Knowledge Tracing (KT), an essential task in ITS, aims to assess the evolution of each student's knowledge state over time based on previous learning interactions and predict their future performance.* Conventional KT models typically assess students' knowledge states based on their interaction history, which usually exhibits data bias[1], *i.e.*, the distribution of question groups (*e.g.*, concepts) is unbalanced. Therefore, KT models often face the issue of **cognitive bias, which usually manifests as cognitive underload on overperformers and cognitive overload on underperformers.** Figure 1(a) illustrates the issue of cognitive overload on underperformers with the example of exercise recommendation. ITS recommends exercises to an underperformer, who

gets 80% incorrect despite a considerable portion of simple questions. However, KT model still assesses that the student is familiar with 20% of the questions, causing the ITS to overestimate the student's abilities and recommend exercises that are difficult to respond to. Meanwhile, cognitive underload will eventually prompt the ITS to recommend low-value exercises to overperformers. Clearly, ITS recommendations, based on the evaluation results of the KT model, that deviate from the current knowledge state of students do not meet the requirements of intelligent education for adaptive learning [28, 35, 42, 48]. What's worse, due to the feedback loop [9, 10], cognitive bias of KT model will be amplified over time (*e.g.*, when an underperformer responds to simple questions incorrectly, ITS may recommend more difficult questions, making it more likely to respond incorrectly) until it reaches a critical point between easy and difficult questions for the student [20, 64], causing the student's real knowledge state to gradually deviate from the model prediction, losing the effectiveness of recommendation, and seriously affecting the students' experience with the ITS.

After scrutinizing the causal relations in the KT model, we attribute cognitive bias to a confounder [46], *i.e.*, the student historical correct rate distribution over question groups. The question features (usually the joint representation of the questions and the concepts) and the student features (*e.g.*, the binary responses to the questions) are usually embedded in the vector chronologically, and then encoded by the KT model to predict the evaluation scores for different concepts. In other words, the KT model evaluates the conditional probability of the student's knowledge state given the question features and student features. From the perspective of causality, the question features and student features can be regarded as the cause of the prediction score, and the KT model performs causal modeling on them. But through the observation of causal relations, we find that the hidden confounder, *i.e.*, the student historical correct rate distribution over question groups, affects both the student representation and the prediction score. Meanwhile, through structured probability modeling, the conventional KT models are affected by the confounder, thereby causing a spurious correlation between the student representation and the prediction score: for questions with higher correct rates, underperformers will get higher prediction scores, even if the students are obviously incorrect, similarly, for questions with lower correct rates, overperformers will get lower prediction scores, even if the students have responded correctly. Figure 1(b) is the empirical evidence of the spurious correlation verified by DKT [49] applied to the assist09 dataset [15]. We calculate the average prediction scores of the model when students respond incorrectly (response=0) and correctly (response=1) across different concepts (Figure 1(b) left). According to the classical test theory [8], we calculate the correct rate of the concept (*i.e.*, the concept difficulty) with average prediction scores $\geq 0.5$ for incorrect responses and with average prediction scores $< 0.5$ for correct responses (Figure 1(b) right). As shown in Figure 1(b) upper right, when the correct rate of almost all concepts is high, although the students respond incorrectly, the model still predicts higher scores, and vice

---
[1]Data bias refers to the over- or under-representation of certain categories, features, or labels relative to others in a dataset. In this work, we focus on measurement bias [17], a common type of data bias, which is consistent with the class imbalance in pattern recognition, *e.g.*, significant differences in the correct rates among different concepts.

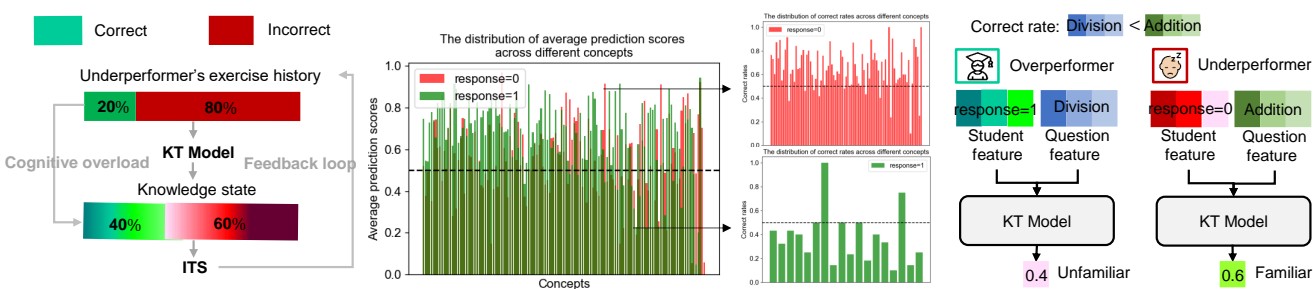

(a) An example of cognitive overload on underperformers.

(b) Empirical evidence of the spurious correlation verified by DKT applied to the assist09 dataset.

(c) The impact of cognitive bias on the knowledge state assessment of different performers.

Figure 1: Illustration of cognitive bias.

versa (Figure 1(b) lower right), which makes the KT model exhibit cognitive bias, undermining the effectiveness of the knowledge state assessment for overperformers and underperformers (see the example in Figure 1(c)).

In order to eliminate the spurious correlation, we propose Disentangled Knowledge Tracing (DisKT), a novel approximate causal model based on causal effects. DisKT models simple and difficult questions separately, thereby modeling the abilities that students are familiar and unfamiliar with, and eliminating the impact of the confounder in student representation within the model. In addition, from Figure 1(b) right, we notice that students make mistakes in questions with extremely high correct rates, and sometimes they can correctly respond to questions with extremely low correct rates. We attribute this to the contradictory psychology (*e.g.*, guessing and mistaking) [7, 12, 16, 33, 72, 75] which are not conducive to the modeling of students' actual knowledge state, which inspires us to design a contradictory attention to shield these factors. Finally, we design a variant of Item Response Theory (IRT) [53, 67], integrating the abilities that students are familiar and unfamiliar with, to enhance the interpretability of the model prediction layer.

In summary, this work contributes in four aspects:

- We analyze the causal relations in the conventional KT model through a causal graph, and reveal the cause of cognitive bias from the perspective of causal probability.
- Based on causal effects, we propose a novel approximate causal model, DisKT, which eliminates the impact of the confounder to alleviate cognitive bias. We also propose a contradiction attention to shield the contradictory psychology (*e.g.*, guessing and mistaking). In addition, we design a variant of IRT to enhance the interpretability of model predictions.
- We construct three datasets with different bias strengths, and design a metric to measure the effectiveness of DisKT in alleviating cognitive bias. In addition, we propose two contradictory metrics to determine the potential of our proposed contradictory attention in shielding guessing and mistaking.
- Extensive experiments on 11 benchmarks from 10 different subjects show that DisKT not only effectively alleviates cognitive bias, but also has superior evaluation accuracy compared to other 14 baselines.

## 2 Related Work

Knowledge Tracing (KT) has been a cornerstone in the development of intelligent tutoring system, enabling personalized education by assessing and predicting students' knowledge states over time [2, 22, 39, 51]. Early attempts at KT, such as the Bayesian Knowledge Tracing (BKT) [12], are based on the Hidden Markov Model, using a binary variable to represent knowledge states. Item Response Theory (IRT) [52] is a factor analysis method designed to model the relationship between students' abilities and their responses by measuring the gap between student ability and question difficulty. With the rise of deep learning, recent studies have utilized deep learning models to address KT issues [2, 49, 57]. DKT [49] first applies LSTM [24] to KT, followed by the introduction of Memory-Augmented Neural Networks (MANN) [55], and DKVMN [73] based on MANN, which uses the key matrix and the value matrix to dynamically store students' mastery of concepts. SKVMN [1] combines the recurrent modeling capability of DKT with the memory network structure of DKVMN. Deep-IRT [68] integrates IRT and DKVMN to make deep learning based KT explainable. Later, GKT [40] uses a graph to model the relationships between knowledge concepts. With the success of attention [5], especially the Transformer [63] in the NLP field, SAKT [41] first introduces self-attention networks to capture the relevance between knowledge concepts and student interactions, followed by state-of-the-art models or frameworks with variant attention structures: AKT [19], DTransformer [70], FoLiBi [27], sparseKT [26]. Meanwhile, some popular training techniques are also applied in KT, such as ATKT [21] and CL4KT [32], which respectively use adversarial training and contrastive learning to enhance student interaction representation. However, despite these studies attempting to address issues in KT and achieving impressive results in evaluation accuracy, there is a lack of comprehensive and reasonable explanation, even Deep-IRT is limited in prediction. In contrast, DisKT is based on causal effect modeling, where the calculation of each interpretable parameter is transparent.

Surprisingly, previous KT research has lacked attention to such an important topic as data bias, and the closest to our DisKT is the Core framework [13]. The Core framework considers that existing models tend to remember answer bias, *i.e.*, the unbalanced distribution of correct and incorrect answers for each question, thus providing a shortcut for achieving good predictive performance in KT and mitigating the answer bias by subtracting the direct causal effect from the total causal effect captured during training in testing [6, 44, 62]. Compared to DisKT, the Core framework has two obvious disadvantages. The Core framework does not recognize the impact of learned bias on different populations, *i.e.*, different cognitive bias exists for overperformers and underperformers, and does not realize that such bias will be amplified. Meanwhile, the Core

framework does not delve into the contradictory psychology [7] in biased data, *e.g.*, guessing and mistaking, and their adverse impact on the real knowledge state. In contrast, DisKT provides a more detailed causal graph, alleviates bias within the model, and designs a contradictory attention to shield contradictory psychology, alleviating bias while also improving evaluation accuracy.

## 3 Methodology

### 3.1 KT Formulation

Formally, we define a set of students $\mathcal{S}$, a set of questions $Q$, and a set of concepts $C$. Historical interactions of a student $s \in S$ are represented as $X_t = \{(q_1, Concept_{q_1}, r_1), (q_2, Concept_{q_2}, r_2), ..., (q_t, Concept_{q_t}, r_t)\}$, where $q_t \in Q$ refers to the question responded to by the student at time $t$, $Concept_{q_t} \subset C$ denotes the set of concepts related to $q_t$, and $r_t \in \{0, 1\}$ indicates the student's response to the question (0 for incorrect, 1 for correct). The aim of KT is to predict the probability $p(r_{t+1} \mid X_t, q_{t+1}, Concept_{q_{t+1}})$ of a student correctly responding to the next question $q_{t+1}$.

### 3.2 Causality Perspective on Cognitive Bias

In this section, we construct a causal graph for the conventional KT model. Based on the causal graph, we probabilistically model the conventional KT model and find that the confounder in student representation is the main culprit leading to cognitive bias. Consequently, we build a counterfactual world to eliminate the influence of the confounder in the real world based on causal effect.

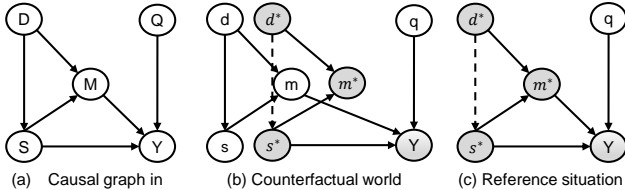

(a) Causal graph in conventional KT
(b) Counterfactual world
(c) Reference situation

**Figure 2: The causal graphs for conventional and counterfactual KT. ∗ denotes the reference values.**

#### 3.2.1 Causal Graph
As shown in Figure 2(a), we use a causal graph to describe the causal relations in KT, which includes five variables: $S, Q, D, M, Y$. Note that we use capital letters (*e.g.*, $D$) to represent variables and lowercase letters (*e.g.*, $d$) to represent specific values of these variables. Specifically,

- $S$ represents student features. For a student, $s = r_{1:t}$ represents the binary response sequence up to time $t$.
- $Q$ represents question features, which is usually a joint representation of the questions and concepts [19, 26].
- $D$ denotes the student historical correct rate distribution over question groups (*e.g.*, concept). Given $n$ concepts $\{c_1, ..., c_n\}$, $d_s = [p_s(c_1), p_s(c_2), ..., p_s(c_n)]$ represents the correct rate distribution of student $s$ across different concepts, where $p_s(c_n)$ refers to the correct rate on concept $c_n$ of student $s$ in history.
- $M$ is the concept-level student representation. A vector $m$ represents the hidden representation of the student, learned by a KT model (*e.g.*, DKT), for different concepts. $m$ is determined only by $s$ and $d$, that is, $m$ can be represented by a function $M(s, d)$ with $s$ and $d$.
- $Y$ with specific value $y \in [0, 1]$ is the prediction score.

- $D \rightarrow S$ indicates that the student historical correct rate distribution over question groups influences the student's representation, tending to overfit unbalanced historical data and exhibiting cognitive bias [54, 64].
- $(D, S) \rightarrow M$ indicates that $D$ and $S$ determine the concept-level student representation.
- $(S, M, Q) \rightarrow Y$ indicates that $S$ affects $Y$ by two paths: the direct path $S \rightarrow Y$, representing the student's actual knowledge state, and the indirect path $S \rightarrow M \rightarrow Y$, which reflects the polarization of the prediction score caused by the bias in the concept-level student representation, *i.e*, simpler question groups are more likely to have higher prediction scores, and more difficult question groups are more likely to be predicted with lower scores.

According to the causal theory [46], $D$ is associated with both $S$ and $Y$, and is a confounder between $S$ and $Y$. Next, through structured probability modeling, we explore how the student historical distribution leads to the polarization of prediction score via biased student representation.

#### 3.2.2 Probability Modeling
Due to the confounder $D$ between $S$ and $Y$, there exists an issue of cognitive bias when the existing KT models predict the conditional probability $P(Y \mid S = s, Q = q)$. Given $S = s$ and $Q = q$, $P(Y \mid S = s, Q = q)$ is formalized as follows:

$$P(Y \mid S = s, Q = q)$$

$$= \frac{\sum_{d \in \mathcal{D}} \sum_{m \in \mathcal{M}} P(Y, s, q, d, m)}{P(s, q)} \tag{1a}$$

$$= \frac{\sum_{d \in \mathcal{D}} \sum_{m \in \mathcal{M}} P(d) P(s \mid d) P(m \mid d, s) P(q) P(Y \mid s, q, m)}{P(s) P(q)} \tag{1b}$$

$$= \sum_{d \in \mathcal{D}} \sum_{m \in \mathcal{M}} P(d \mid s) P(m \mid d, s) P(Y \mid s, q, m) \tag{1c}$$

$$= \sum_{d \in \mathcal{D}} P(d \mid s) P(Y \mid s, q, M(d, s)) \tag{1d}$$

$$= P(d_s \mid s) P(Y \mid s, q, M(d_s, s)), \tag{1e}$$

where $\mathcal{D}$ and $\mathcal{M}$ are the sample spaces of $D$ and $M$, respectively. Eq. (1a), Eq. (1b), and Eq. (1c) are derived from the total probability rule, causal graph, and Bayes formula, respectively. And $m$ is determined by certain $d$ and $s$, so we get Eq. (1d). We only study the sample space $d_s$ of student $s$, thus obtaining Eq. (1e).

From Figure 2(a) and Eq. (1e), we find that $D$ not only affects $S$ but affects $Y$ through $M(d_s, s)$, causing a spurious correlation: for questions in simpler or more difficult question group (*e.g.*, concept $c_n$), the prediction scores are higher or lower, *i.e.*, the high or low prediction scores are caused by the student historical distribution rather than the questions themselves. In Eq. (1e), a higher or lower correct rate $p_s(c_n)$ in $d_s$ will make $M(d_s, s)$ show a better or worse knowledge state of concept $c_n$, and then increase or decrease the prediction scores of questions in $c_n$ through $P(Y \mid s, q, M(d_s, s))$. This ultimately leads to the polarization of prediction scores for questions in easy and difficult concepts, *i.e.*, the cognitive bias towards overperformers and underperformers.

#### 3.2.3 Counterfactual
Counterfactual technology is a method of estimating causal effects by considering events that may occur under different conditions and analyzing "how the results would change if the situation were different" [50]. As shown in Figure 2(b),

we construct a counterfactual world: $D$ does not affect $Y$ through $S$ but only affects $Y$ through $M$, that is, the counterfactual can estimate how much the prediction score would be if $D$ had no effect on $S$. The key to the counterfactual is to intervene causally on $S$, also called the do-operator [45, 47, 66], *i.e.*, $do(S = s^*)$ forcibly cuts off the edge $D \rightarrow S$ in Figure 2(a), replaces $s$ in Eq. (1e) with $s^*$, and obtains $P(Y \mid do(S = s^*), Q = q) = p(d)p(s^*)P(Y \mid s^*, q, M(d, s^*))$, where $d$ **can be considered a constant distribution.**

**3.2.4 Causal Effect** In causal effects, the Total Effect ($TE$) of $S = s$ on $Y$ denotes the change in $Y$ caused by the $S$ when it changes from the reference value $s^*$ in Figure 2(c) to the expected value $s$ in Figure 2(a). Given $Q = q$, the $TE$ of $S = s$ on $Y$ is formalized as:

$$TE = Y_{s,m,q} - Y_{s^*,m^*,q}, \tag{2}$$

where $Y_{s^*,m^*,q}$ represents the reference state of $Y$ when $S = s^*$, and $S$ is not affected by $D$. Typically, $TE$ can be decomposed into $TE = NDE + TIE$, where $NDE$ and $TIE$ respectively represent the natural direct effect and total indirect effect [6, 44, 62].

Specifically, given $Q = q$, the $NDE$ of $M = m$ on $Y$ refers to the change in the prediction score $Y$ when the $M$ changes from the reference value $m^*$ to the expected value $m$ and $do(S = s^*)$ is enforced. The $NDE$ is formalized as follows:

$$NDE = Y_{s^*,m,q} - Y_{s^*,m^*,q}, \tag{3}$$

where $Y_{s^*,m,q}$ is the prediction score in the counterfactual world, and $S$ is not affected by $D$ and $M$ remains unchanged (as shown in Figure 2(b)).

Therefore, given $Q = q$, the $TIE$ of $S = s$ can be obtained by subtracting $NDE$ from $TE$:

$$TIE = TE - NDE = Y_{s,m,q} - Y_{s^*,m,q}. \tag{4}$$

Therefore, the $TIE$ of $S = s$ on $Y$ is the change in the $Y$ caused by the $S$ when it changes from the reference value $s^*$ to the expected value $s$ without affecting the $M$.

In Eq. 4, $NDE$ estimates how much the prediction score would be in the counterfactual world if the KT model only had the student historical distribution and did not track the student's knowledge state. Intuitively, $TIE$ represents the final prediction score, which reduces the $NDE$ of the student historical distribution [13, 64]. Therefore, **the prediction score for underperformers on questions of high correct rate would be largely suppressed, conversely, the prediction score for overperformers on questions of high correct rate would be liberated.**

## 3.3 Disentangled Knowledge Tracing

Through the analysis of causal effects, the key to eliminating the influence of the confounder lies in how to causally intervene on $S$ so that the student representation after $do(S = s^*)$ only represents the student historical distribution. In addition, the trouble caused by the factual and counterfactual inference of the traditional causal model is also a problem worth considering [13, 64, 65, 74]. To this end, DisKT we propose is an approximate causal model. Considering that the student historical distribution is fundamentally indicating that students are familiar with simple concepts but not good at difficult concepts, DisKT models the responses of incorrect and correct responses separately along with the concepts, roughly classifying the concepts responded to correctly as simple concepts, and the concepts responded to incorrectly as difficult concepts, thus

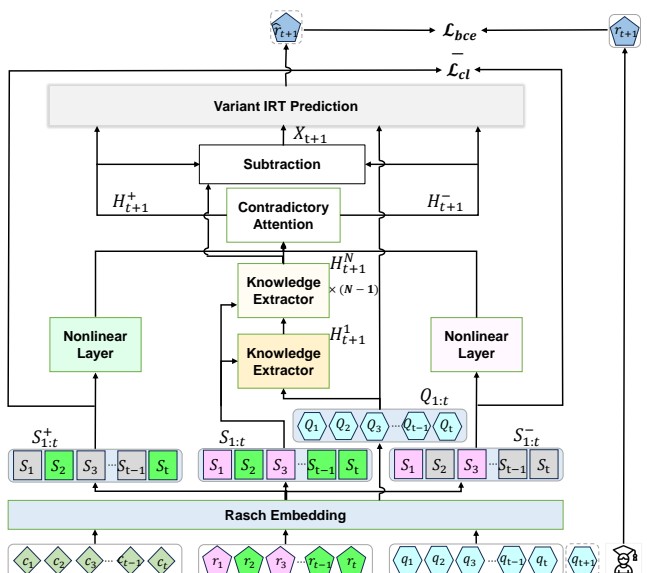

**Figure 3: The architecture of the Disentangled Knowledge Tracing model (DisKT).**

modeling the students' familiarity and unfamiliarity abilities. **Due to separate modeling, it is difficult to track knowledge of either side, thus achieving that the student representation after intervention approximately represents the student historical distribution.** In addition, to avoid double causal inference, DisKT executes the process of causal intervention within the model, and approximates $M$ by assigning contradictory attention weights considering the contradictory psychology (*e.g.*, guessing and mistaking) [7, 12, 16, 33, 72, 75] of the factual student representation to both sides, thereby reducing the burden of re-inference.

The architecture of DisKT is shown in Figure 3, with details as follows.

**3.3.1 Rasch Embedding** KT models often describe multiple concepts as a single concept, *i.e.*, $c_t = Concept_{q_t}$, and due to data sparsity, they use concepts instead of questions as the subject of assessment [19, 32, 49]. We use the Rasch model in psychology [19, 36, 38, 52] to construct the $t$-th embeddings of question (*i.e.*, $Q_t$) and interaction (*i.e.*, $S_t$):

$$Q_t = c_{c_t} + d_{q_t} \cdot \mu_{c_t}, S_t = e_{(c_t, r_t)} + d_{q_t} \cdot v_{(c_t, r_t)},$$
$$e_{(c_t, r_t)} = c_{c_t} + r_{r_t}, v_{(c_t, r_t)} = c_{c_t} + g_{r_t}, \tag{5}$$

where $c_{c_t} \in \mathbb{R}^d$ is the embedding of concept $c_t$, $d_{q_t} \in \mathbb{R}$ is a scalar representing the difficulty of question $q_t$, $\mu_{c_t} \in \mathbb{R}^d$ summarizes the variation of questions containing concept $c_t$, $r_{r_t} \in \mathbb{R}^d$ is the embedding of response $r_t$, $g_{r_t} \in \mathbb{R}^d$ is the variation embedding of response $r_t$, $e_{(c_t, r_t)} \in \mathbb{R}^d$ and $v_{(c_t, r_t)} \in \mathbb{R}^d$ are the embedding and variation embedding of the concept-response interaction $(c_t, r_t)$. $d$ denotes the dimension of these embeddings.

Therefore, given the student's historical interaction sequence $\{q_{1:t}, c_{1:t}, r_{1:t}\}$, the factual embeddings of questions (*i.e.*, $Q_{1:t}$) and interactions (*i.e.*, $S_{1:t}$) are represented as follows:

$$S_{1:t}, Q_{1:t} = \text{Rasch Embedding}(q_{1:t}, c_{1:t}, r_{1:t}). \tag{6}$$

Through artificial intervention, that is, in order to obtain the correct interaction embeddings, DisKT masks the elements corresponding to the incorrect response positions in $q_{1:t}, c_{1:t}$ and $r_{1:t}$, and vice versa, the obtained counterfactual interaction embeddings $S_{1:t}^+$ and $S_{1:t}^-$ are:

$$S_{1:t}^+ := \text{Rasch Embedding}(r_{1:t} \cdot q_{1:t}, r_{1:t} \cdot c_{1:t}, mask + (1 - mask) \cdot r_{1:t}), \tag{7}$$

$$S_{1:t}^- := \text{Rasch Embedding}((1 - r_{1:t}) \cdot q_{1:t}, (1 - r_{1:t}) \cdot c_{1:t}, mask \cdot r_{1:t}),$$

where $mask$ is the mask value (*e.g.*, 2) of the response sequence, while the question and concept sequences are masked by 0.

**3.3.2 Knowledge Extractor** In order to effectively encode the embeddings of questions and interactions, the knowledge extractor employs $N$ Transformer encoders [63]. For the first encoder, the knowledge extractor takes the question and interaction embeddings $Q_{1:t}$ and $S_{1:t}$ as input and outputs the knowledge state $H_{t+1}^1$ extracted from the current questions and interactions:

$$\begin{cases} H_{t+1}^1 = \text{LN}(\text{Dropout}(\text{Res}(\text{FFN}(S = \text{Multihead}^1)))), \\ \text{FFN}(S) = \text{GeLU}(SW^1 + b^1)W^2 + b^2, \\ \text{Multihead}^1 = \text{Concat}(\text{head}_1^1, \cdots, \text{head}_h^1)W_h^1, \\ \text{head}_i^1 = \text{Attention}(Q = Q_{1:t}^{/h}, K = Q_{1:t}^{/h}, V = S_{1:t}^{/h}), \\ \text{Attention}(Q, K, V) = \text{Concat}(\textbf{zero}, \text{Softmax}(\frac{QK^T}{\sqrt{d/h}})\,[1:,:])V, \end{cases} \tag{8}$$

where LN, Dropout, Res, FFN refer to layer normalization [4], dropout technique [58], residual connection [23] and fully-connected feed-forward, respectively, $W^1 \in \mathbb{R}^{d \times d}, W^2 \in \mathbb{R}^{d \times d}, b^1 \in \mathbb{R}^d, b^2 \in \mathbb{R}^d$ are learnable parameters and $\text{GeLU}(\cdot)$ is the activation function, $h$ is the number of attention heads (*e.g.*, 2) and $W_h^1 \in \mathbb{R}^{d \times d}, Q_{1:t}^{/h}$ and $S_{1:t}^{/h}$ represent splitting the $d$ dimensions of $Q_{1:t}$ and $S_{1:t}$ into $h$ parts, respectively, and $\textbf{zero} \in \mathbb{R}^d$ is a zero vector indicating that the historical interaction before the first question is not available. Eq. 8 can be abbreviated as

$$H_{t+1}^1 = \text{Encoder}(Q = Q_{1:t}, K = Q_{1:t}, V = S_{1:t}), \tag{9}$$

so the output of the last encoder is

$$H_{t+1}^N = \text{Encoder}(Q = H_{t+1}^{N-1}, K = H_{t+1}^{N-1}, V = S_{1:t}). \tag{10}$$

The design of the knowledge extractor enables DisKT to summarize the performance of students over multiple time scales and extract comprehensive knowledge.

**3.3.3 Contradictory Attention** Considering the burden of re-reasoning and the impact of contradictory psychology (*e.g.*, guessing and mistaking), the contradictory attention we designed assigns the knowledge learned by the factual student representation from the knowledge extractor to the student representations in the counterfactual world, which previously perform feature extraction through a nonlinear layer (*e.g.*, FFN). Meanwhile, it shields the weights of the contradictory psychology through a selective Softmax function (Softmax*). Finally, it forms the knowledge states $H_{t+1}^+$ and $H_{t+1}^-$ representing familiar and unfamiliar abilities in the counterfactual world:

$$\begin{cases} H_{t+1}^+ = \text{Attention}^*(Q = H_{t+1}^N, K = H_{t+1}^N, V = \text{FFN}(S_{1:t}^+)), \\ H_{t+1}^- = \text{Attention}^*(Q = H_{t+1}^N, K = H_{t+1}^N, V = \text{FFN}(S_{1:t}^-)), \\ \text{Attention}^*(Q, K, V) = \text{Concat}(\textbf{zero}, \text{Softmax}^*(X = \frac{QK^T}{\sqrt{d/h}})[1:, \\ \quad :])V, \\ \text{Softmax}^*(X) = \text{Softmax}(\textbf{one} - exp(CV(c_{1:t}, r_{1:t}), d) \cdot \text{Softmax}(X)), \\ CV(c_t, r_t) = \mathbb{1}(\max(\lambda_t, \beta)(1 - r_t + (2r_t - 1) \cdot \text{diff}(c_t)) < \alpha_t^2), \end{cases} \tag{11}$$

where $\textbf{one} \in \mathbb{R}^{d \times d}$ is a vector of ones, $exp(\cdot, d)$ represents column expansion by $d$ dimensions, $CV(c_{1:t}, r_{1:t})$ represents the contradictory variable sequence determined by $c_{1:t}$ and $r_{1:t}$, and $\mathbb{1}(\cdot)$ denotes the indicator function. $\lambda_t$ is a random value, representing the probability that the student is not affected by the contradictory psychology at time $t$, and the smaller it is, the more likely it is to be affected by the contradictory psychology. $\beta$ is the lower bound of $\lambda_t$ (*e.g.*, 0.1), preventing the dominant position of the uncertain $\lambda_t$. $\alpha_t$ is a determined value, representing the degree threshold of a student's contradictory psychology at time $t$. $\text{diff}(c_t)$ refers to the correct rate obtained from the training set through $c_t$ according to the classical test theory [8].

The form of $CV(c_t, r_t)$ is intuitive: the more difficult $c_t$ is, the more likely the student is to guess; the simpler $c_t$ is, and the student responses incorrectly, the more likely the student is to experience a psychology of mistaking. $\alpha_t$ can be determined as follows:

$$\begin{cases} \alpha_t = \gamma, & \text{if } t = 1, \\ \alpha_t = \sqrt{\frac{\sum_{i=1}^{t-1}(\max(\lambda_i, \beta)(1 - r_i + (2r_i - 1) \cdot \text{diff}(c_i)))}{t-1}}, & \text{else,} \end{cases} \tag{12}$$

where $\gamma$ (*e.g.*, 0.2) is the initial value of the student's contradiction. As can be seen from Eq. 12, if a student has more and more severe contradictory psychology in the past, the contradiction threshold should be higher, and vice versa.

**3.3.4 Variant IRT Prediction** Since the prediction scores range from 0 to 1, DisKT subtracts the $NDE$ from Eq. 4 in terms of features:

$$X_{t+1} = H_{t+1} - (H_{t+1}^+ + H_{t+1}^-). \tag{13}$$

Then, DisKT explicitly indicates the questions to be predicted and generates final prediction scores through an $MLP$:

$$\hat{r} = \sigma(\text{ReLU}([(X_{t+1} - d_{q_{1:t}}) \oplus (H_{t+1}^+ - H_{t+1}^-) \oplus Q_{1:t}]W_1 + b_1)W_2 + b_2), \tag{14}$$

where $\oplus$ denotes the concatenation operation. $W_1 \in \mathbb{R}^{3d \times d}, W_2 \in \mathbb{R}^{d \times 1}, b_1 \in \mathbb{R}^d, b_2 \in \mathbb{R}^1$ are learnable parameters in the $MLP$. $\sigma(\cdot)$ denotes the sigmoid function and $\text{ReLU}(\cdot)$ is the activation function. $d_{q_{1:t}}$ can be obtained from Eq. 5. Eq. 14 not only considers the student's overall ability and the question difficulty, but also integrates the abilities that the student is familiar and unfamiliar with, making the prediction more interpretable.

**3.3.5 Model Training** DisKT applies a binary cross-entropy loss to directly optimize the assessment of knowledge state:

$$\mathcal{L}_{bce} = -\sum_{i=1}^{t}(r_i\log\hat{r}_i + (1 - r_i)\log(1 - \hat{r}_i)). \tag{15}$$

                                                                                                                     

**Table 1: Comparison of DisKT and 14 KT models on 11 datasets. The averages across five test folds are reported. Best results in bold, next best underlined. %Improv. denotes the relative performance improvement achieved by DisKT over the strongest baseline. \* and \*\* indicate that the improvements over the strongest baseline are statistically significant, with $p$ <0.05 and $p$ <0.01, respectively. A model with ✓ indicates that it is interpretable.**

| Dataset | Metric | DKT | DKVMN | SKVMN | Deep-IRT✓ | GKT | SAKT | AKT | ATKT | CL4KT | CoreKT✓ | DTransformer | simpleKT | FoLiBiKT | sparseKT | DisKT✓ | %Improv. |
|---|---|---|---|---|---|---|---|---|---|---|---|---|---|---|---|---|---|
| assist09 | AUC↑ | 0.7591 | 0.7570 | 0.7434 | 0.7566 | 0.7484 | 0.7348 | 0.7705 | 0.7543 | 0.7597 | 0.7415 | 0.7508 | 0.7709 | 0.7710 | 0.7670 | **0.7923**\*\* | 2.76% |
| | ACC↑ | 0.7166 | 0.7172 | 0.7084 | 0.7180 | 0.7127 | 0.7000 | 0.7192 | 0.7171 | 0.7194 | 0.7020 | 0.7042 | 0.7209 | 0.7165 | 0.7092 | **0.7275** | 0.92% |
| | RMSE↓ | 0.4333 | 0.4333 | 0.4391 | 0.4334 | 0.4374 | 0.4442 | 0.4349 | 0.4348 | 0.4339 | 0.4436 | 0.4505 | 0.4372 | 0.4356 | 0.4396 | **0.4298** | -0.81% |
| algebra05 | AUC | 0.7780 | 0.7713 | 0.6260 | 0.7698 | 0.7806 | 0.7493 | 0.7932 | 0.7624 | 0.7864 | 0.7579 | 0.7694 | 0.7874 | 0.7923 | 0.7806 | **0.8033**\*\* | 1.27% |
| | ACC | 0.7893 | 0.7896 | 0.7570 | 0.7886 | 0.7889 | 0.7800 | 0.7938 | 0.7882 | 0.7940 | 0.7643 | 0.7877 | 0.7927 | 0.7925 | 0.7856 | **0.8009**\*\* | 0.87% |
| | RMSE | 0.3854 | 0.3862 | 0.4212 | 0.3865 | 0.3851 | 0.3942 | 0.3811 | 0.3899 | 0.3833 | 0.4038 | 0.3906 | 0.3842 | 0.3818 | 0.3875 | **0.3786**\* | -0.66% |
| algebra06 | AUC | 0.7598 | 0.7685 | 0.6294 | 0.7663 | 0.7578 | 0.7330 | 0.7740 | 0.7426 | 0.7714 | 0.7509 | 0.7391 | 0.7695 | 0.7731 | 0.7694 | **0.7846**\*\* | 1.37% |
| | ACC | 0.7934 | 0.7989 | 0.7762 | 0.7977 | 0.7874 | 0.7845 | 0.7953 | 0.7878 | 0.7968 | 0.7710 | 0.7871 | 0.7923 | 0.7967 | 0.7940 | **0.8033**\*\* | 0.6% |
| | RMSE | 0.3823 | 0.3784 | 0.4089 | 0.3794 | 0.3851 | 0.3921 | 0.3797 | 0.3891 | 0.3798 | 0.3946 | 0.3892 | 0.3814 | 0.3787 | 0.3804 | **0.3755** | -0.77% |
| statics | AUC | 0.7611 | 0.7596 | 0.6692 | 0.7515 | 0.7528 | 0.7304 | 0.7999 | 0.7421 | 0.7783 | 0.7894 | 0.7690 | 0.7888 | 0.7988 | 0.7887 | **0.8086**\*\* | 1.09% |
| | ACC | 0.7643 | 0.7688 | 0.7211 | 0.7661 | 0.7657 | 0.7560 | 0.7761 | 0.7658 | 0.7691 | 0.7598 | 0.7732 | 0.7723 | 0.7794 | 0.7775 | **0.7883** | 1.14% |
| | RMSE | 0.4027 | 0.4019 | 0.4359 | 0.4059 | 0.4017 | 0.4148 | 0.3889 | 0.4130 | 0.3966 | 0.3990 | 0.3966 | 0.3924 | 0.3894 | 0.3909 | **0.3823**\* | -1.7% |
| ednet | AUC | 0.6589 | 0.6657 | 0.6531 | 0.6656 | 0.6569 | 0.6499 | 0.7003 | 0.6544 | 0.6651 | 0.6697 | 0.6978 | 0.7048 | 0.6995 | 0.7006 | **0.7384**\*\* | 4.8% |
| | ACC | 0.6289 | 0.6346 | 0.6259 | 0.6337 | 0.6193 | 0.6240 | 0.6592 | 0.6243 | 0.6349 | 0.6284 | 0.6559 | 0.6573 | 0.6582 | 0.6557 | **0.6863**\*\* | 4.11% |
| | RMSE | 0.4771 | 0.4756 | 0.4784 | 0.4759 | 0.4788 | 0.4809 | 0.4746 | 0.4797 | 0.4759 | 0.4762 | 0.4799 | 0.4730 | 0.4756 | 0.4727 | **0.4592**\*\* | -2.86% |
| prob | AUC | 0.7159 | 0.7192 | 0.7038 | 0.7190 | 0.7098 | 0.7100 | 0.7376 | 0.7062 | 0.7213 | 0.7293 | 0.7354 | 0.7265 | 0.7270 | 0.7437 | **0.7731**\*\* | 3.95% |
| | ACC | 0.6786 | 0.6888 | 0.6768 | 0.6900 | 0.6806 | 0.6818 | 0.7015 | 0.6849 | 0.6877 | 0.6889 | 0.6922 | 0.6971 | 0.6974 | 0.7057 | **0.7215**\*\* | 2.24% |
| | RMSE | 0.4543 | 0.4520 | 0.4564 | 0.4521 | 0.4555 | 0.4562 | 0.4491 | 0.4585 | 0.4524 | 0.4537 | 0.4518 | 0.4498 | 0.4522 | 0.4483 | **0.4353**\*\* | -2.9% |
| linux | AUC | 0.7421 | 0.7470 | 0.6991 | 0.7441 | 0.7402 | 0.7348 | 0.8225 | 0.7532 | 0.7580 | 0.7837 | 0.8211 | 0.8221 | 0.8216 | 0.8249 | **0.8622**\*\* | 4.52% |
| | ACC | 0.7625 | 0.7643 | 0.7535 | 0.7634 | 0.7612 | 0.7595 | 0.7977 | 0.7657 | 0.7674 | 0.7576 | 0.7979 | 0.7968 | 0.7945 | 0.7983 | **0.8152**\*\* | 2.12% |
| | RMSE | 0.4042 | 0.4029 | 0.4151 | 0.4038 | 0.4067 | 0.4075 | 0.3741 | 0.4010 | 0.4002 | 0.4047 | 0.3742 | 0.3746 | 0.3756 | 0.3734 | **0.3601**\*\* | -3.56% |
| comp | AUC | 0.7239 | 0.7170 | 0.6631 | 0.7150 | 0.7132 | 0.7082 | 0.7986 | 0.7256 | 0.7243 | 0.7420 | 0.7988 | 0.8000 | 0.7979 | 0.7964 | **0.8324**\*\* | 4.05% |
| | ACC | 0.8037 | 0.8017 | 0.7991 | 0.8015 | 0.7914 | 0.8000 | 0.8203 | 0.8048 | 0.8040 | 0.7825 | 0.8217 | 0.8191 | 0.8196 | 0.8197 | **0.8264**\* | 0.57% |
| | RMSE | 0.3788 | 0.3808 | 0.3899 | 0.3813 | 0.3878 | 0.3833 | 0.3589 | 0.3784 | 0.3805 | 0.3915 | 0.3582 | 0.3585 | 0.3592 | 0.3591 | **0.3510**\* | -2.01% |
| database | AUC | 0.7490 | 0.7531 | 0.6924 | 0.7498 | 0.7497 | 0.7419 | 0.8263 | 0.7546 | 0.7587 | 0.7839 | 0.8184 | 0.8272 | 0.8253 | 0.8367 | **0.8769**\*\* | 4.8% |
| | ACC | 0.8336 | 0.8340 | 0.8278 | 0.8330 | 0.8327 | 0.8317 | 0.8485 | 0.8346 | 0.8328 | 0.7883 | 0.8478 | 0.8497 | 0.8500 | 0.8531 | **0.8690**\*\* | 1.86% |
| | RMSE | 0.3530 | 0.3522 | 0.3644 | 0.3528 | 0.3538 | 0.3553 | 0.3310 | 0.3518 | 0.3523 | 0.3830 | 0.3328 | 0.3301 | 0.3302 | 0.3266 | **0.3090**\*\* | -5.39% |
| spanish | AUC | 0.8029 | 0.8081 | 0.7277 | 0.8032 | 0.8114 | 0.7950 | 0.8391 | 0.8047 | 0.8202 | 0.8273 | 0.8170 | 0.8408 | 0.8399 | 0.8395 | **0.8529**\*\* | 1.44% |
| | ACC | 0.7443 | 0.7508 | 0.6952 | 0.7461 | 0.7496 | 0.7417 | 0.7745 | 0.7545 | 0.7550 | 0.7613 | 0.7513 | 0.7734 | 0.7735 | 0.7718 | **0.7847** | 1.45% |
| | RMSE | 0.4156 | 0.4145 | 0.4482 | 0.4177 | 0.4155 | 0.4236 | 0.3968 | 0.4186 | 0.4119 | 0.4053 | 0.4108 | 0.3963 | 0.3962 | 0.3959 | **0.3872**\*\* | -2.2% |
| slepemapy | AUC | 0.6861 | 0.6989 | 0.6473 | 0.6935 | 0.6539 | 0.6709 | 0.7258 | 0.6952 | 0.7097 | 0.7135 | 0.7217 | 0.7269 | 0.7230 | 0.7255 | **0.7632**\*\* | 4.99% |
| | ACC | 0.7780 | 0.7789 | 0.7786 | 0.7782 | 0.7844 | 0.7739 | 0.7835 | 0.7790 | 0.7857 | 0.7238 | 0.7865 | 0.7868 | 0.7826 | 0.7876 | **0.7944**\*\* | 0.86% |
| | RMSE | 0.3991 | 0.3978 | 0.4046 | 0.3998 | 0.4009 | 0.4050 | 0.3906 | 0.3983 | 0.3938 | 0.4205 | 0.3892 | 0.3889 | 0.3916 | 0.3903 | **0.3808**\*\* | -2.08% |

In addition, in order to expedite model convergence and ensure that the model learns two different types of abilities, familiar and unfamiliar, DisKT introduces an additional regularization term to constrain model learning:

$$\mathcal{L}_{cl} = \|S_{1:t}^{+} - S_{1:t}^{-}\|. \tag{16}$$

Therefore, the final objective function of DisKT is:

$$\mathcal{L}_{DisKT} = \mathcal{L}_{bce} - \mathcal{L}_{cl}. \tag{17}$$

## 4 Experiments

We conduct extensive experiments, aiming to answer the following four research questions to demonstrate the effectiveness of DisKT:

- **RQ1:** How does DisKT perform compare to various state-of-the-art KT models?
- **RQ2:** How does DisKT alleviate cognitive bias compared to the most advanced KT models?
- **RQ3:** How effective is DisKT in shielding guessing and mistaking?
- **RQ4:** What are the impacts of the components (*e.g.*, contradictory attention) on DisKT?

## 4.1 Experimental Setup

**4.1.1 Datasets** We evaluate the performance of DisKT on 11 public datasets: assist09, algebra05, algebra06, statics, ednet, prob, linux, comp, database, spanish, slepemapy. The introduction and detailed processing of the datasets can be found in Appendix A, and Table 3 presents the statistics of the processed datasets.

**4.1.2 Baselines** We compare DisKT with 14 state-of-the-art models as follows: DKT [49], DKVMN [73], SKVMN [1], Deep-IRT [68], GKT [40], SAKT [41], AKT [19], ATKT [21], CL4KT [32], CoreKT [13], DTransformer [70], simpleKT [36], FoLiBiKT [27], sparseKT [26]. Their introductions can be found in Appendix B.

**4.1.3 Implementation Details** We adopt 5-fold cross-validation and folds are split based on the students. 10% of the training set is used for validation, which is not only used for parameter tuning but for early stopping strategy, that is, if AUC does not improve within 10 epochs, the training is halted. We focus on the most recent 100 interactions per student, as this recent information is crucial for future predictions [32]. The models are optimized by Adam [30] with the following settings: the batch size is 512, the learning rate is set to 0.001, the dropout rate is 0.05, and the embedding dimension is 64. All models are trained in PyTorch [43] on a Linux server with two GeForce RTX 3090s. Following the previous works [27, 32, 56], the evaluation metrics include Area Under the ROC Curve (AUC), Accuracy (ACC) and Root Mean Square Error (RMSE). Our code and datasets are available at https://anonymous.4open.science/r/DisKT.

## 4.2 Comparison with SOTA (RQ1 & RQ2)

**4.2.1 Overall Performance** *w.r.t.* **Accuracy** Table 1 presents the evaluation performance of the compared models in terms of AUC, ACC, and RMSE. Overall, our DisKT consistently outperforms other baselines on all metrics across all datasets. The main observations are as follows:

- DisKT has quite substantial evaluation performance on almost all datasets. Specifically, in terms of AUC, DisKT's average relative improvement over the strongest baseline on all datasets is 3.2%.

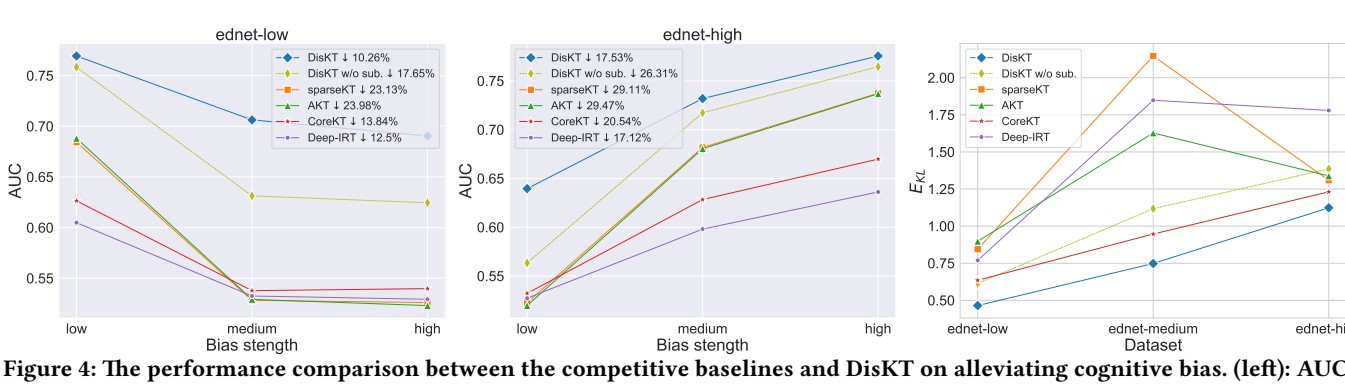

**Figure 4: The performance comparison between the competitive baselines and DisKT on alleviating cognitive bias. (left): AUC performance changes of DisKT and baselines optimized with different bias strengths on ednet, tested with ednet-low and ednet-high respectively. (right): $E_{KL}$ scores of several representative models on three datasets with different bias strengths.**

These impressive results demonstrate the effectiveness of our causal modeling, enabling DisKT to eliminate the influence of confounder (*i.e.*, student historical distribution) in the student representation, and therefore, the model can generate correct cognition for underperformers and overperformers, thereby improving the accuracy of the evaluation.

- Interpretable models like Deep-IRT and CoreKT show poorer performance in biased data due to sacrificing performance for interpretability. For instance, CoreKT instantiated based on AKT generally performs worse than AKT. In contrast, DisKT eliminates the spurious correlation between user representation and prediction score within the model, thereby alleviating the amplification of cognitive bias, bringing performance improvements while also enhancing interpretability.

- Compared to other neural network structures, the attention mechanism, especially Transformer, significantly affects the performance of KT models. This is consistent with the research results in [36, 37]. This gap is more pronounced on larger datasets (the statistical information of the datasets is shown in Table 3), indicating the superiority of the attention mechanism for large-scale real-world datasets, *i.e.*, it is expert in capturing long-term dependencies, thus it can extract rich information from large-scale data as [19, 36] described. And DisKT significantly outperforms other Transformer-based models in larger datasets, proving that our DisKT can better exploit the powerful potential of Transformer.

**4.2.2    Performance on Alleviating Cognitive Bias**  Due to data bias, the KT model tends to exhibit cognitive bias in different student populations: overperformers can't solve difficult questions, while underperformers tend to get simple questions right. To further evaluate the effectiveness of DisKT in alleviating cognitive bias, we conduct experiments on synthetic data. Specifically, according to the classical test theory [8], we build three datasets of different bias strengths based on ednet according to the frequency of correct responses ($< 60\%$, $60\%\sim 80\%$, and $\geq 80\%$): ednet-low, ednet-medium, and ednet-high. These datasets are consistent with the settings in Section 4.1.3. We choose two interpretable models, Deep-IRT and CoreKT, and two best-performing models, AKT and sparseKT, as baselines for studying cognitive bias. We test the models optimized by the three datasets with ednet-low and ednet-high, respectively, and their AUC performance changes are shown in Figure 4 left. We make the following observations. 1) The AUC performance of all

models trained by endnet-high and tested with the endnet-low has decreased, which indicates that models that have only seen high-accuracy, *i.e.*, simple questions, overload the cognition of underperformers. Similarly, the AUC performance of all models trained by endnet-low and tested with the endnet-high has also decreased, reflecting the model's cognition underload for overperformers. 2) Compared to other strong competitors, DisKT effectively alleviates the two cognitive biases while maintaining optimal evaluation performance. In contrast, even though interpretable models like Deep-IRT and CoreKT achieve good results, this comes at the cost of sacrificing evaluation performance. 3) Cognitive bias greatly affects the evaluation performance of the baselines, even rendering their evaluation ineffective, *i.e.*, AUC performance is around 0.5, while DisKT still maintains AUC performance above 0.6. However, the AUC performance of DisKT drops significantly after removing the causal effect (DisKT $w/o$. sub.), indicating the correctness of our modeling based on causal effect.

However, some metrics like AUC can only indirectly reflect the impact of cognitive bias on the model. In KT, there is a lack of a direct measure that can reflect the amplification degree of cognitive bias in the model. To fill this gap, inspired by [60], we have designed a calibration metric $E_{KL}$, which measures the gap between the actual and predicted correct rate distributions when the model makes incorrect predictions. The calculation method can be found in Appendix C. Higher $E_{KL}$ scores suggest a more serious issue of cognitive bias. The $E_{KL}$ scores of several representative models on three datasets with different bias strengths are shown in Figure 4 right. We can see that, regardless of the bias strength of the dataset, DisKT always maintains the lowest $E_{KL}$ scores. Meanwhile, CoreKT achieves sub-optimal results due to its mitigation of answer bias. Deep-IRT, due to its simple structure, cannot adapt to datasets with larger bias strengths. AKT and sparseKT learn data biases but are unable to suppress the amplification of cognitive bias within the model, leading to the worst results. This further proves the effectiveness of DisKT in alleviating cognitive bias.

## 4.3    In-depth Analysis (RQ3 & RQ4)

**4.3.1    Effect of Shielding Guessing and Mistaking**  In Section 1, we find that there existing the contradictory psychology (*e.g.*, guessing and mistaking) in the students' biased data. We have provided empirical evidence of guessing and mistaking using DKT on the

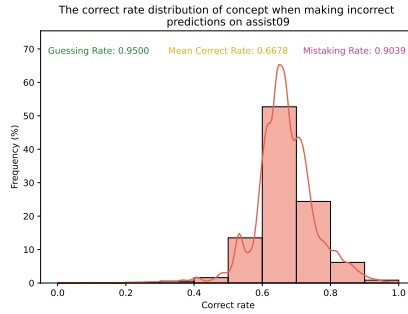

**Figure 5: Empirical evidence of guessing and mistaking verified by DKT applied to the assist09 dataset.**

assist09 dataset. Specifically, we have analyzed the correct rate distribution of concept when DKT makes incorrect predictions. We consider concepts with a correct rate of less than or equal to 0.3 as difficult concepts and those with a correct rate of greater than or equal to 0.7 as easy concepts. We then calculate the proportion of correct responses by students for difficult concepts (Guessing Rate) and the proportion of incorrect responses for easy concepts (Mistaking Rate). The results, as shown in Figure 5, indicate that the Guessing Rate and Mistaking Rate are surprisingly high at 0.9500 and 0.9039, respectively. This suggests that contradictory psychology (*e.g.*, guessing and mistaking) can easily have a negative impact on the assessment of the real knowledge state. Intuitively, the Guessing Rate and Mistaking Rate measure the adverse effects of guessing and mistaking on the model, respectively, with lower values indicating that the model is better at shielding the impact of these inevitable psychological factors. Table 2 presents the contradictory metrics (Guessing Rate and Mistaking Rate) of DisKT and several representative baselines on three datasets with different bias strengths. We note that Deep-IRT achieves good results in the Guessing Rate on all datasets, indicating that purely introducing question difficulty helps reduce the adverse impact of guessing. Meanwhile, DisKT achieves the best results in Mistaking Rate on all datasets while generally outperforming other baselines in Guessing Rate. However, the results of DisKT without the contradictory attention (DisKT *w/o.* con.) are generally pessimistic, confirming the effectiveness of our designed contradictory attention in shielding guessing and mistaking.

**4.3.2    Ablation Study**  We have constructed five variants of DisKT to explore the impact of different components on DisKT, as shown in Figure 6. Specifically, in addition to the previously mentioned "w/o. sub." and "w/o. con.", "w/o. IRT" removes the variant IRT module, "w. nor. IRT" uses a normal IRT module, and "w/o. $loss_{cl}$" omits the loss function $loss_{cl}$. The following observations are made: (1) "w/o. con." shows a similar degree of performance decline across both datasets, highlighting the importance of contradiction attention in shielding guessing and mistaking and effectively tracking

**Table 2: Performance comparison in terms of shielding guessing and mistaking.**

| Dataset | Metric | Deep-IRT | CoreKT | AKT | sparseKT | DisKT w/o. con. | DisKT |
|---|---|---|---|---|---|---|---|
| ednet-low | Guessing Rate↓ | **0.7372** | 0.7968 | 0.9633 | 0.9577 | 0.9417 | 0.7515 |
| | Mistaking Rate↓ | 0.9129 | 0.8292 | 0.8631 | 0.9295 | 0.8710 | **0.7406** |
| ednet-medium | Guessing Rate | **0.3627** | 0.8000 | 0.9853 | 0.9855 | 0.9559 | 0.7778 |
| | Mistaking Rate | 0.9557 | 0.9097 | 0.8298 | 0.8304 | 0.9241 | **0.6904** |
| ednet-high | Guessing Rate | 0.3796 | 0.5612 | 0.4397 | 0.4427 | 0.3985 | **0.2797** |
| | Mistaking Rate | 0.9596 | 0.9233 | 0.9314 | 0.9304 | 0.9851 | **0.8895** |

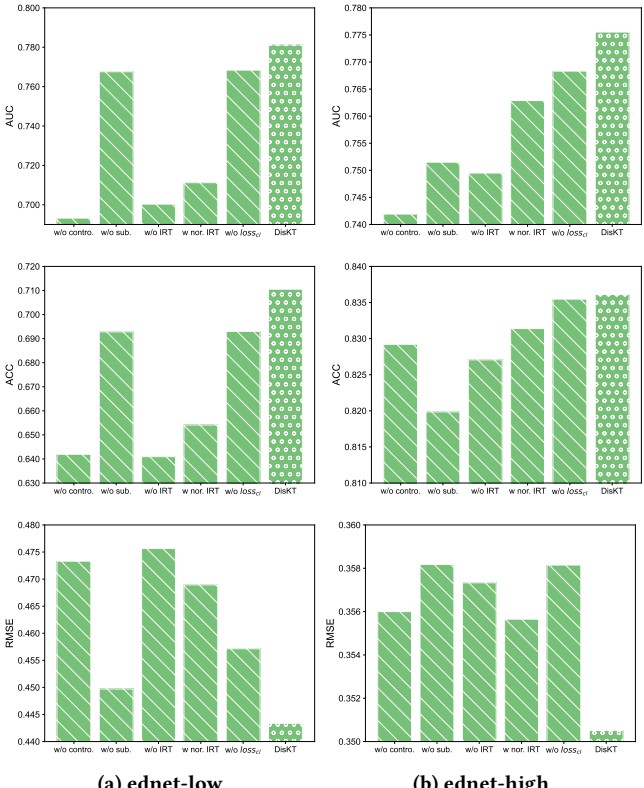

**(a) ednet-low**          **(b) ednet-high**

**Figure 6: Ablation study on ednet-low and ednet-high.**

knowledge states. (2) "w/o. sub." exhibits a slight performance decline on ednet-low and a sharp decline on ednet-high, indicating that DisKT, which models based on causal effects, is more adaptable to data with high bias. (3) "w. nor. IRT" experiences a significant performance decline, and "w/o. IRT" even more so, emphasizing not only the effectiveness of the IRT module but also the advantages of our proposed the variant IRT module. (4) "w/o. $loss_{cl}$" shows a slight performance decline, indicating that the regularization term $loss_{cl}$ accelerates convergence while also effectively learning the abilities of familiarity and unfamiliarity.

## 5    Conclusion and Future Work

In this work, we elucidate that cognitive bias within KT models stems from the confounder from a causal perspective. In response, we propose the DisKT model based on causal effect, effectively nullifying the impact of the confounder in student representation. Moreover, DisKT incorporates a contradiction attention to shield the contradictory psychology (*e.g.*, guessing and mistaking) in the students' biased data. Meanwhile, we innovate a variant of IRT to enhance the interpretability of model predictions. Our findings, supported by rigorous experiments across 11 benchmarks and 3 synthesized datasets, reveal that DisKT not only significantly alleviates cognitive bias but also surpasses 14 baseline models in terms of evaluation accuracy.

The avenues of future work include *i*) further exploration of educational psychology, such as forgetting, *ii*) investigation of the critical points between simple and difficult questions for different students and *iii*) discovery of more fine-grained causal relations.

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

## A Datasets

We evaluate the performance of DisKT on 11 public datasets:

- **assist09**[2] [15]: The assist09 dataset, composed of math exercises, is collected from the ASSISTment intelligent tutoring system in the school year 2009-2010 and is widely used as a standard benchmark in KT research.
- **algebra05, algebra06**[3] [59]: The algebra05 and algebra06 datasets come from the KDD Cup 2010 EDM Challenge, containing detailed step-level student responses to algebra questions.
- **statics**[4] [31]: The statics dataset is a collection of records from a college-level engineering statics course at Carnegie Mellon University during the Fall semester of 2011.
- **ednet**[5] [11]: The ednet dataset, collected by the multi-platform AI tutoring service Santa, stands as the largest publicly released interactive educational system dataset to date.
- **prob, comp, linux, database**[6] [25]: The prob, comp, linux, database datasets are collected from the Programming Teaching Assistant platform, specifically from course exercises in Probability and Statistics, Computational Thinking, Linux System, and Database Technology and Application.
- **spanish**[7] [34]: The spanish dataset is from middle-school students practicing Spanish exercises, including translations and applications of basic skills like verb conjugation, over a 15-week semester.
- **slepemapy**[8] [? ]: The slepemapy dataset originates from slepemapy.cz, an online platform dedicated to the adaptive practice of geography facts.

Following the data preprocessing approach in [18], we exclude students with fewer than five interactions and all interactions involving nameless concepts. **Since a question may involve multiple concepts, we convert the unique combinations of concepts within a single question into a new concept.** The statistical information after processing is shown in Table 3. It's noted that we randomly sample 5000 students from three large datasets, ednet, comp and slepemapy.

**Table 3: Statistics of 11 datasets. "#concepts*" denotes the total number of concepts after converting multiple concepts into a new concept.**

| Datasets | #students | #questions | #concepts | #concepts* | #interactions |
|---|---|---|---|---|---|
| assist09 | 3,644 | 17,727 | 123 | 150 | 281,890 |
| algebra05 | 571 | 173,113 | 112 | 271 | 607,014 |
| algebra06 | 1,138 | 129,263 | 493 | 550 | 1,817,450 |
| statics | 333 | 1,223 | N/A | N/A | 189,297 |
| ednet | 5,000 | 12,117 | 189 | 1,769 | 676,276 |
| prob | 512 | 1,054 | 247 | 247 | 42,869 |
| comp | 5,000 | 7,460 | 445 | 445 | 668,927 |
| linux | 4,375 | 2,672 | 281 | 281 | 365,027 |
| database | 5,488 | 3,388 | 291 | 291 | 990,468 |
| spanish | 182 | 409 | 221 | 221 | 578,726 |
| slepemapy | 5,000 | 2,723 | 1,391 | 1,391 | 625,523 |

---

[2]https://sites.google.com/site/assistmentsdata/home/2009-2010-assistment-data/skill-builder-data-2009-2010
[3]https://pslcdatashop.web.cmu.edu/KDDCup
[4]https://pslcdatashop.web.cmu.edu/DatasetInfo?datasetId=507
[5]https://github.com/riiid/ednet
[6]https://github.com/wahr0411/PTADisc
[7]https://github.com/robert-lindsey/WCRP
[8]https://www.fi.muni.cz/adaptivelearning/?a=data

## B Baselines

We compare DisKT with 14 state-of-the-art models as follows:

- **DKT** [49]: DKT is the first model that employs Recurrent Neural Networks (RNNs) to solve the KT task. Over the past few years, it has been widely used as a standard baseline in KT research.
- **DKVMN** [73]: DKVMN leverages a dual-matrix approach to refine KT, using a static key matrix for mapping interconnections among concepts and a dynamic value matrix for real-time updates of a student's knowledge state.
- **SKVMN** [1]: SKVMN integrates the recurrent modeling capabilities of DKT with the advanced memory network structure of DKVMN, enhancing the representation and tracking of students' knowledge states over time.
- **Deep-IRT** [68]: Deep-IRT provides a detailed understanding of student learning trajectories and the difficulty of concepts, integrating the DKVMN for feature extraction with the psychometric insights of IRT.
- **GKT** [40]: GKT redefines the KT task by modeling the knowledge structure as a graph, transforming it into a node-level classification challenge.
- **SAKT** [41]: SAKT employs a self-attention mechanism within a Transformer architecture to dynamically weigh past learning interactions, capturing long-term dependencies and the relevance among concepts and historical interactions.
- **AKT** [19]: AKT accounts for the learner's tendency to forget over time within its monotonic attention framework by integrating embeddings inspired by the Rasch model.
- **ATKT** [21]: ATKT applies these perturbations to the original student interaction sequences, utilizing an attention-based LSTM framework.
- **CL4KT** [32]: CL4KT introduces a novel contrastive learning framework for KT, aiming to enhance representation learning by distinguishing between similar and dissimilar learning histories.
- **CoreKT** [13]: CORE framework enhances KT by addressing answer bias through a causality perspective. It differentiates between total and direct causal effects of questions on student responses to mitigate bias, improving the accuracy of tracing students' knowledge states. We introduce CORE with AKT, namely CoreKT.
- **DTransformer** [70]: DTransformer revolutionizes KT by diagnosing learner's knowledge states from question-level mastery using a novel architecture. It employs Temporal and Cumulative Attention (TCA) mechanisms for dynamic analysis and a contrastive learning-based algorithm for stable knowledge state tracking.
- **simpleKT** [36]: simpleKT introduces a simple but tough-to-beat baseline for KT, focusing on simplicity and robust performance across diverse KT datasets.
- **FoLiBiKT** [27]: FoLiBi, leveraging the forgetting-aware linear bias concept, innovatively addresses the challenge of modeling forgetting behavior in KT by introducing a linear bias mechanism. We introduce FoLiBi with AKT, namely FoLiBiKT.
- **sparseKT** [26]: sparseKT introduces a k-selection module designed to select items that achieve the highest attention scores, integrating two distinct sparsification strategies: soft-thresholding sparse attention and top-K sparse attention.

## C  Evaluation Metric

In the experiments, we design a calibration metric $E_{KL}$ to measure the amplification degree of cognitive bias within KT models, which is defined by the following equation.

$$E_{KL} = \sum_c \frac{p_c}{\sum_c p_c} \frac{\log \frac{p_c}{\sum_c p_c}}{\frac{q_c}{\sum_c q_c}},$$

$$p_c = \frac{\sum_{i=1}^t \mathbb{1}(c_i = c, r_i = 1, (f_i < 0.5) = r_i)}{\sum_{i=1}^t \mathbb{1}(c_i = c, (f_i < 0.5) = r_i)},$$

$$q_c = \frac{\sum_{i=1}^t f_i \cdot \mathbb{1}(c_i = c, (f_i < 0.5) = r_i)}{\mathbb{1}(c_i = c, (f_i < 0.5) = r_i)},$$

$$f_i = f(q_{1:i-1}, c_{1:i-1}, r_{1:i-1}),$$

where $\mathbb{1}(\cdot)$ is the indicator function. $f$ is the model to be evaluated. $c_i$ and $r_i$ represent the $i$-th concept and its response, respectively. There are a total of $t$ interactions.

