# OpenReview forum: "Disentangled Knowledge Tracing for Alleviating Cognitive Bias"
_ACM.org/TheWebConf/2025/Conference — WWW 2025 Poster_

### Official Review · Reviewer_8CdA · 2024-11-14

**Novelty:** 4
**Technical Quality:** 4

**Review:**

1. Brief summary：
This paper explores how KT models in ITS can accurately assess students‘ learning states, pointing out that traditional KT models are prone to cognitive bias due to data bias, such as uneven distribution of question groups (e.g., concepts). This bias not only leads to cognitive underloading in overperformers and overloading in underperformers, but also exacerbates in practice recommendations. The impact of the distribution of students' historical correct rate on the prediction of learning performance was found to be the main issue, and the DisKT model was proposed. The model eliminates the influence of confounder by causally modeling students' familiar and unfamiliar abilities separately. At the same time, DisKT introduces a contradiction attention mechanism to mitigate the effects of guessing and mistaking, and incorporates a variant of IRT to enhance the interpretability of mode. Experimental results on 11 baseline and 3 synthetic datasets show that DisKT significantly mitigates cognitive bias and outperforms 14 baseline models in assessment accuracy. However, the paper still has issues regarding overall logicality, research background, metric design, and other aspects, and requires further revision.

2. Quality:
Pros：
(1) The paper is well-structured, with precise language and no discernible grammatical issues. Most mathematical symbols are defined clearly and without ambiguity, facilitating a thorough comprehension of the content.
(2) This paper uncovers cognitive bias inherent in KT models and presents a comprehensive analysis of their origins. Through an examination of causal relationships, it elucidates the potential influence of students' historical correct rate distributions on model predictions, thereby shedding light on the mechanism behind cognitive bias formation. This finding offers a fresh perspective and approach for addressing cognitive biases in KT models.
(3) The paper proposes the DisKT model, which effectively eliminates cognitive bias by separating modeling the concepts that students are familiar and unfamiliar with. In addition, the model introduces a contradiction attention mechanism to shield the influence of psychological factors such as guessing and mistaking, thus enhancing the robustness and accuracy of the model. Meanwhile, the DisKT model integrates variants of IRT, making the model prediction results more interpretable.
(4) The paper conducts extensive experiments on 11 baseline datasets and 3 synthetic datasets with different bias strengths, and the results show that the DisKT model performs well in mitigating cognitive bias, and at the same time outperforms the other 14 baseline models in terms of assessment accuracy. These ample experimental results validate the effectiveness of the DisKT model and demonstrate its superior performance in knowledge tracing tasks.

Cons:
(1) While the paper presents the novel issue of cognitive bias, many concepts are not clearly defined; for instance, the definitions of "underperformers" and "overperformers" are vague, and the causes of cognitive bias are not sufficiently explained. Furthermore, the content of Figure 1(b) is unclear and lacks precise description, making it difficult for readers to fully understand the research problem.
(2) The paper validates the model through extensive experiments, which show that DisKT outperforms most baseline models in terms of overall performance. However, DisKT mainly focuses on cognitive bias caused by data bias, and only one dataset, ednet, was used in the validation process, which was divided into three subsets for the experiments. Therefore, it is recommended that experiments on more datasets be added or that the $E_{KL}$ metric be considered as a complementary metric to the overall experiments in Table 1 to more fully demonstrate the effectiveness of DisKT in addressing cognitive bias.
(3) The paper proposes using a contradiction attention mechanism to address the issues of student guessing and mistaking, and defines two new metrics, the guessing rate and the mistaking rate, to validate that DisKT resolves this issue. However, the rationale behind the calculation of these metrics remains questionable. For example, while concepts with a correct rate of less than or equal to 0.3 are considered difficult, some students who have truly mastered these concepts (rather than guessing) are still included in the calculation. The same concern applies to the error rate.
(4) DisKT uses a variant of IRT to enhance the model's interpretability, but it only demonstrates the effectiveness of this variant in improving overall model performance through ablation experiments, without explicitly clarifying where the interpretability is reflected. It is suggested that the authors further elaborate on the specific ways in which the IRT variant contributes to the model's interpretability, for instance, by using visual analysis or concrete examples to show how the variant makes the model's decision-making process more transparent and understandable.
(5) The paper focuses on the key issues of cognitive bias, contradictory psychology, and interpretability but fails to effectively establish the intrinsic connections between them. The structure appears piecemeal and lacks deeper integration. This issue is also present in the model design, where parts lack organic cohesion in theoretical foundation and methodological application. As a result, the article struggles to thoroughly reveal the interconnections and interactions among these three elements, which somewhat weakens the overall logical coherence.

3. Clarity:
Overall, this paper has a good format and contains no obvious grammatical errors, making it easy to understand the content. However, there are still clarity issues that the authors are advised to address:
(1) Line 48-49: “the distribution of question groups (e.g., concepts) is unbalanced” is suggested to add a reference explanation.
(2) Line 52-53: The latter text mentions underperformers and overperformers several times, but does not explain how to distinguish between underperformers and overperformers, it is suggested to add a detailed explanation.
(3) Line 64-66: “However, KT model still assesses that the student is familiar with 20% of the questions, causing the ITS to overestimate the student's abilities and recommend exercises that are difficult to respond to.” Regarding this point, although the student answers 80% of the questions incorrectly, the KT model still assesses that the student is familiar with 20% of the questions. This is merely an objective reflection of the data and does not necessarily indicate the cause of the ITS overestimating the student's abilities. Therefore, the causal relationship here needs further exploration and clarification.
(4) Lines 81-83: It is recommended that the authors include data or figures to demonstrate that the distribution of students' historical correct rate is a form of data bias, which leads to cognitive bias in the KT model.
(5) Line 109-112: This section is vague and does not clearly explain the specific meaning of Figure 1(b) (right), nor does it specify how the correct rate is calculated.
(6) Lines 117-128: In Figure 1(b), the arrows indicate that Figure 1(b) right is derived from the two separated results of Figure 1(b) left. However, the vertical axis of Figure 1(b) left represents the average predicted score, while the vertical axis of Figure 1(b) right represents correct rate. The meaning of the two figures is unclear, and the description of the figure in the introduction is also ambiguous.
(7) Line 133: The content of Figure 1(c) is not explained in detail.
(8) Lines 137-138: The phrase "abilities that students are familiar and unfamiliar with" is mentioned repeatedly here and in the surrounding context. Does "abilities" refer to knowledge concepts or groups of knowledge concepts? If so, it is recommended that the authors explicitly specify "knowledge concepts."
(9) Line 208: It is recommended to use "FoLiBiKT" consistently with the context instead of "FoLiBi."
(10) Line 297: In the relationship (S, M, Q) → Y, three paths should be included, but the third path, Q → Y, is not explained.
(11) Line 399: Change “propose” to “proposed”.
(12) Line 430: Change “familiarity and unfamiliarity” to “familiar and unfamiliar”.
(13) Line 578: Eq. 15 is missing a parenthesis.
(14) Line 713: "student historical distribution" should be consistent with the context. It is recommended that the authors change it to "the student historical correct rate distribution."
(15) Line 769: Change “endnet-high” to “ednet-high”, change “endnet-high”  to  “ednet-high”, the same as line 773.
(16) Line 825-826: What does "the correct rate distribution of concept" mean? It is recommended to provide a detailed explanation and distinguish it from the student's correct rate.
(17) Line 829-832: Is the calculation of the guessing rate and mistaking rate reasonable? For instance, although concepts with a correct rate of less than or equal to 0.3 are considered difficult, some students might not be guessing but have truly mastered those concepts, yet they are still included in the calculation. The same issue applies to the calculation of the mistaking rate.
(18) Line 1185: An invalid reference.

4. Originality:
(1) Analyzing Cognitive Bias from a Causal Perspective: This paper is the first to analyze the causes of cognitive bias in KT models from a causal perspective, revealing the potential impact of students' historical correct rate distributions on model predictions. This provides a new approach to addressing cognitive bias.
(2) Introduction of the Novel Approximate Causal Model, DisKT: The paper introduces a novel approximate causal model, DisKT, which separates the modeling of students' familiarity and unfamiliarity with knowledge. It also incorporates a contradiction attention mechanism and a variant IRT prediction module, effectively mitigating cognitive bias, enhancing model robustness, and improving prediction interpretability.
(3) Construction of Datasets with Different Bias Intensities: The paper constructs datasets with varying levels of bias intensity and designs new metrics to evaluate the model's effectiveness in alleviating cognitive bias. This provides a valuable reference for related research.

5. Significance:
The paper provides an in-depth analysis of the roots of cognitive bias in KT models and proposes the DisKT model based on causal relationships. The DisKT model effectively mitigates cognitive bias by separately modeling students' familiarity with familiar and unfamiliar knowledge, and by integrating a contradiction attention mechanism with a variant IRT module. This significantly enhances the model's evaluation accuracy and reliability. The research not only advances the field of knowledge tracing and lays the foundation for building more effective KT systems, but also offers strong support for personalized learning and the improvement of educational quality. By more accurately assessing students' knowledge states, the DisKT model can help ITS recommend more suitable learning resources and paths, thus improving learning efficiency and outcomes. However, the paper also has some potential shortcomings, with issues related to the overall logicality, the clarity of the research background, the sufficiency of experiments, and the design of evaluation metrics, which require further revisions.

**Questions:**

(1) The $E_{KL}$ metric you proposed directly measures the amplification of cognitive bias in the model. Would it be possible to apply this metric across all baseline models and datasets to more clearly demonstrate the effectiveness of DisKT in mitigating cognitive bias?
(2) How do you handle extreme cases where students perform poorly but are not merely guessing when calculating the guessing rate and mistaking rate? Is there a more refined way to define "difficult" concepts or mistaking rate to improve the fairness and accuracy of the model?

**Reviewer Confidence:**

3: The reviewer is confident but not certain that the evaluation is correct

**Scope:**

3: The work is somewhat relevant to the Web and to the track, and is of narrow interest to a sub-community

---

### Official Review · Reviewer_wgxe · 2024-11-24

**Novelty:** 7
**Technical Quality:** 7

**Review:**

Updated Review:

The authors have alleviated my concerns about computational complexity through a detailed analysis of inference time and memory consumption. The clarification provided alleviates concerns regarding computational requirements, as the model does not utilize mLSTM. Furthermore, the efficacy of DisKT in cold-start scenarios was demonstrated through its effectiveness with abbreviated interaction sequences. This supplementary data enhances the paper's practical contributions and reflects careful attention to reviewers' comments.

Main Review:

In this paper, DisKT (Disentangled Knowledge Tracing) is introduced as a new method that tackles cognitive bias in knowledge tracing systems by using causal analysis. The role is at the juncture of educational data mining and causal inference, which is significant for the user modeling and personalization track at the WWW conference.
The main focus of the paper is on analyzing theories and providing practical solutions for cognitive bias in knowledge tracing. Initially, the authors pinpoint the confounding effect of students' historical accuracy rate distribution as the main factor contributing to cognitive bias through causal analysis. Next, they suggest a two-step plan: a technique to improve student representation using causal effects, and a mechanism to address psychological factors such as guessing and mistaking. Their unique method of separately modeling students' known and unknown skills is particularly innovative, successfully removing the confounding factor in student representation.
The work shows great technical depth from a methodological point of view. The causal analysis is thorough, with a precise definition of the issue and a structured building of theoretical basis. The execution demonstrates a high level of focus on details, especially in the development of the contradiction attention mechanism and the variant IRT module. Yet, the paper could be improved by delving deeper into the analysis of computational complexity and memory needs, particularly in light of the matrix operations used in the model.
The thorough empirical assessment was done on 11 benchmarks and 3 synthesized datasets featuring various levels of bias. The experimental setup adheres to good practices, including detailed comparisons with 14 baseline models using various evaluation metrics. The ablation studies successfully confirm specific parts, especially the conflicting attention mechanism and sequence improvement module. The results show substantial enhancements in accuracy measures and addressing bias, along with a thorough examination of how the model performs with various student groups.
Yet, there are multiple limitations that should be taken into account.
1.	Although the theoretical basis is solid, the paper could improve by examining how well the model works in various educational settings and with different student groups.
2.	Further elaboration is needed on the computational complexity and memory demands of matrix operations in mLSTM.
3.	The way cold-start situations are managed, especially for brand new students or concepts, could be improved with more detailed attention.
4.	Empirical validation could enhance the adaptation of conflicting psychological thresholds.
The presentation is mostly clear, featuring a well-organized argument and suitable application of mathematical symbols. The target audience can understand the technical content, but adding more examples or simpler explanations, especially in the methodology, would be helpful. The authors' arguments are well-supported by the causal graphs and empirical evidence figures.
Looking at the bigger picture, this work has made several important contributions to the field.
1.	The analysis provided is distinct in its approach to examining cognitive bias in knowledge tracing.
2.	It presents a practical answer that enhances both precision and comprehensibility.
3.	It shows the effective application of causal inference in educational technology.
4.	New metrics are available for evaluating cognitive bias and paradoxical psychological effects.
The authors offer thorough experimental settings and implementation details for reproducibility. The mathematical equations are well-defined and thorough. Further exploration of hyperparameter sensitivity and initialization strategies would be helpful.
The paper shows a high level of scientific rigor in both its theoretical framework and empirical testing. The writers methodically construct their argument by analyzing causes, developing models, and conducting thorough evaluations. By comparing with various baselines and conducting thorough ablation studies, they present compelling proof of the efficacy of their method.
The work focuses on important challenges in online education platforms from a practical perspective. The suggested solution seems feasible with current systems, but there should be more discussion on deployment considerations. Enhancements in managing students, whether they are active or passive, show promise for making a real difference in the world.

Academic Language and Referencing Analysis:

The academic tone and scholarly format of the paper show strengths as well as room for improvement. The writers consistently use the correct technical terms in the manuscript, effectively incorporating domain-specific language when discussing causal inference, knowledge tracing, and contradictory psychology. The text demonstrates a well-organized academic framework and a logical flow, presenting technical ideas with the necessary mathematical precision. Yet, various parts could use more accurate and formal wording. Sometimes, the introduction and discussion sections use informal language that takes away from the scholarly style - for example, phrases like "what's worse" (p.2) and "greatly outnumber" could be substituted with more academic terms. The methodology section has redundant phrases that could be revised for better clarity and impact.
The method of referencing displays consistency in quality. The authors display a consistent citation style and exhibit a strong grasp of fundamental knowledge by referencing literature, specifically in the related work section. A well-rounded bibliography consists of influential pieces and modern research. Yet, some crucial assertions could use more empirical evidence with citations. Specifically, assertions regarding MOOC dropout rates and conflicting psychological impacts need more support from current research. The technical base, particularly in terms of mLSTM execution and causal inference methods, requires a more thorough referencing of recent research from 2023-2024. The list of references shows formatting inconsistencies, with some entries missing important bibliographic details like page numbers or volume numbers.
Improving technical documentation involves providing more detailed descriptions of how things are implemented. Although the mathematical equations are usually easy to understand, it would be helpful to include more references to important works in the methodology section, especially when discussing causal intervention and contradiction attention mechanisms. The validity and reliability of the proposed metrics could be better established through more detailed statistical analysis and comparison with existing measures.

**Questions:**

1.	Computational Requirements:
Can you give me a thorough examination of DisKT's computational complexity and memory needs, specifically focusing on the matrix operations in mLSTM and the contradiction attention mechanism?
2.	Causal Assumptions:
To what extent can we depend on the causal assumptions and intervention plans in diverse educational fields and with various student populations? Have these assumptions been tested in a range of educational settings?
3.	Cold-start Handling:
How well does the model handle brand new students or concepts? Do we need a certain number of interactions for consistent performance?
4.	Contradiction Thresholds:
Could you elaborate on how the contradiction threshold αt is modified in response to diverse student behaviors and learning patterns? What empirical data underpins the selected strategy for adaptation?
5.	Scalability:
What is the relationship between the model's performance and the growing numbers of students, exercises, and concepts? What limitations need to be considered when deploying on a large scale?

**Ethics Review Description:**

The study utilizes common public datasets and strives to enhance educational results by enhancing knowledge tracing. Student privacy and the impact on learning are adequately addressed in the paper. The causal intervention approach aims to help both students who participate and those who do not.

**Reviewer Confidence:**

4: The reviewer is certain that the evaluation is correct and very familiar with the relevant literature

**Scope:**

4: The work is relevant to the Web and to the track, and is of broad interest to the community

---

### Official Review · Reviewer_hUFL · 2024-11-26

**Novelty:** 5
**Technical Quality:** 4

**Review:**

The paper appears to tackle an important problem in Knowledge Tracing (KT), and the authors propose a novel approach called Disentangled Knowledge Tracing (DisKT). However, the methodology, particularly the causal modeling and contradiction attention mechanism, is highly technical and complex, making it challenging to fully assess without a deeper familiarity with the field. While the experimental results seem thorough, the clarity of the explanations is insufficient for a comprehensive understanding of how each component functions and contributes to the overall model.

The paper's technical depth makes it difficult for someone less experienced in the nuances of causal inference and KT models to follow. Key concepts like the confounder effects, contradiction psychology, and causal graph-based modeling are not sufficiently explained for broader accessibility. The reliance on mathematical formulations without intuitive descriptions limits understanding.

While the work claims originality in addressing cognitive bias through causal modeling and attention mechanisms, it is hard to verify the extent of novelty due to insufficient familiarity with prior research. The use of datasets with varying bias strengths and custom metrics for bias evaluation does suggest innovation, but the practical implications and scalability of these contributions remain unclear.

The significance of the work is somewhat evident from its application to Intelligent Tutoring Systems (ITS). However, the impact of DisKT on real-world educational settings is not explicitly discussed or demonstrated. This makes it hard to gauge how transformative the work might be beyond academic benchmarks.

**Questions:**

1) Are the proposed metrics for bias evaluation applicable outside the experimental datasets used in the paper?
2) How robust is the model when applied to new or unseen educational data?
3) How does DisKT perform when deployed in a real-world ITS environment? Are there any practical use cases or deployments planned?

**Reviewer Confidence:**

2: The reviewer is willing to defend the evaluation, but it is likely that the reviewer did not understand parts of the paper

**Scope:**

3: The work is somewhat relevant to the Web and to the track, and is of narrow interest to a sub-community

---

### Official Review · Reviewer_YrSs · 2024-12-01

**Novelty:** 4
**Technical Quality:** 4

**Review:**

This paper discusses the challenges faced by traditional Knowledge Tracing (KT) models in Intelligent Tutoring Systems (ITS) due to data bias, particularly the unbalanced distribution of question groups (e.g., concepts), which causes cognitive bias. This results in cognitive underload for overperforming students and overload for underperforming ones, further exacerbated by exercise recommendations from ITS. The main cause of this issue is identified as the confounder effect of students' historical performance across question groups. To address this, the authors propose a new model called Disentangled Knowledge Tracing (DisKT), which separates students' familiar and unfamiliar abilities, thereby mitigating the impact of confounders. DisKT also introduces a contradiction attention mechanism to account for psychological biases like guessing and mistakes in students’ responses. Moreover, the model enhances interpretability by incorporating a variant of Item Response Theory. Experimental results demonstrate that DisKT significantly reduces cognitive bias and outperforms 14 baseline models in accuracy across various datasets.

Reasons To Accept:

1.The topic studied in this paper has practical applications.

2.The method is a bit little novel.

3.This work is experimented on many datasets and baselines to validate the effectiveness of the methods proposed in this paper.


Reasons To Reject:

1.There are too many separate symbols in the methods, which makes the paper hard to understand.

2.The correspondence between each of the points presented in the methods section and the motivation presented in the introduction section needs to be further clarified.

3.This method emphasizes the influence of students' historical accuracy distribution on students' representation and prediction, However, this may rely too much on historical performance of students' actual learning progress, ignore students' progress in current learning tasks or short-term memory, and may limit the adaptability and flexibility of the model.

**Questions:**

1.How does the Equation 12 come from?

2.How does the counterfactual KT work? What is the relation between the counterfactual KT and disentangled knowledge tracing?

3.I notice that most baselines are before 2024. Why don’t you compare with the latest baselines in 2024?

4.What is the base model of this work in experiments?

5.Are there any reasons other than uneven distribution of concepts that could affect performance?

**Reviewer Confidence:**

2: The reviewer is willing to defend the evaluation, but it is likely that the reviewer did not understand parts of the paper

**Scope:**

3: The work is somewhat relevant to the Web and to the track, and is of narrow interest to a sub-community

---

### Official Review · Reviewer_Zn3H · 2024-12-03

**Novelty:** 6
**Technical Quality:** 6

**Review:**

### Summary
This paper introduces the Disentangled Knowledge Tracing (DisKT) method, which aims to address cognitive bias in existing Knowledge Tracing (KT) models used in Intelligent Tutoring Systems (ITS). Specifically, inspired by the fact that conventional KT models suffer from cognitive bias due to unbalanced data distributions, the proposed method separates students' familiar and unfamiliar abilities based on causal effects and includes a contradiction attention mechanism to shield contradictory psychology in students' biased data. Experimental results on various benchmarks demonstrate that DisKT significantly alleviates cognitive bias and outperforms existing models in evaluation accuracy.

---

### Strengths
* The authors provide a detailed explanation of the problem of cognitive bias in Knowledge Tracing (KR) models along with intuitive experiment results.
* The approach to tackle the cognitive bias issue in KT models seems sound and reasonable, which includes a causal analysis (that disentangles familiar and unfamiliar abilities) and a contradiction attention mechanism (that helps mitigate the impact of guessing and mistaking in student responses).
* The experimental results are comprehensive, covering multiple datasets and comparisons with various recent models.

---

### Weaknesses
* I am not an expert in this domain and I have a hard time understanding the problem setup and the proposed method. It would be beneficial if the authors could include some real-world examples of the target problem and how the proposed method acts on it.
* It is not clear how the proposed DisKT can enhance the interpretability of the model prediction.
* I don't see a clear connection between this work and the theme of the Web. It would be better if the authors could discuss some rationale for it.

**Questions:**

Similar to the last weaknesses above, I am wondering the reason why the authors state that the proposed DisKT can enhance the interpretability is due to the modeling of the causal effect.

**Reviewer Confidence:**

2: The reviewer is willing to defend the evaluation, but it is likely that the reviewer did not understand parts of the paper

**Scope:**

2: The connection to the Web is incidental, e.g., use of Web data or API